# Association Study of the *SLC1A2* (rs4354668), *SLC6A9* (rs2486001), and *SLC6A5* (rs2000959) Polymorphisms in Major Depressive Disorder

**DOI:** 10.3390/jcm11195914

**Published:** 2022-10-07

**Authors:** Patryk Rodek, Małgorzata Kowalczyk, Jan Kowalski, Aleksander Owczarek, Piotr Choręza, Krzysztof Kucia

**Affiliations:** 1Department and Clinic of Adult Psychiatry, Faculty of Medical Sciences, Medical University of Silesia, Ziolowa 45, 40-635 Katowice, Poland; 2Department of Medical Genetics, Faculty of Pharmaceutical Sciences, Medical University of Silesia, Jedności 8, 41-200 Sosnowiec, Poland; 3Health Promotion and Obesity Management Unit, Department of Pathophysiology, Faculty of Medical Sciences in Katowice, Medical University of Silesia, Medyków 15, 40-752 Katowice, Poland; 4Department of Statistics, Faculty of Pharmaceutical Sciences in Sosnowiec, Medical University of Silesia, 41-200 Katowice, Poland

**Keywords:** MDD, depression, glutamate system, excitatory amino acid transporter 2, glycine transporter 1, glycine transporter 2, polymorphism

## Abstract

The membrane excitatory amino acid transporter 2 (EAAT2), encoded by *SLC1A2,* is responsible for the uptake and redistribution of synaptic glutamate. Glycine modulates excitatory neurotransmission. The clearance of synaptic glycine is performed by glycine transporters encoded by *SLC6A9* and *SLC6A5*. Higher synaptic glycine and glutamate levels could enhance the activation of NMDA receptors and counteract the hypofunction of glutamate neurotransmission described in major depressive disorder (MDD). The aim of the study was to assess whether polymorphisms of *SCL1A2* (rs4354668), *SCL6A5* (rs2000959), and *SCL6A9* (rs2486001) play a role in the development of MDD and its clinical picture in the Polish population. The study group consisted of 161 unrelated Caucasian patients with MDD and 462 healthy unrelated individuals for control. Polymorphisms were genotyped with PCR-RLFP assay. We observed that the frequency of genotype CC and allele C of the *SLC1A2* polymorphism rs4354668 was twice as high in the MDD group as in control. Such differences were not detected in *SLC6A5* and *SLC6A9* polymorphisms. No statistically significant association of the studied SNPs (Single Nucleotide Polymorphisms) on clinical variables of the MDD was observed. The current study indicates an association of polymorphism rs4354668 in *SCL1A2* with depression occurrence in the Polish population; however, further studies with larger samples should be performed to clarify these findings.

## 1. Background

Depression is a commonly occurring, serious, and recurrent disorder linked to diminished role functioning and quality of life, medical morbidity, and mortality, with more than 264 million people affected worldwide [1,2]. The heritability of depression is significant, with around 35% of the risk associated with genetic predisposition [3]. Genome-wide association studies (GWASs) have identified thousands of loci implicated in the pathogenesis of MDD (each having a small effect size) and confirmed a highly polygenic architecture of the disease [4]. Candidate gene studies have long been performed, but their positive findings are now being questioned, as 18 well-studied MDD candidate genes (e.g., *BDNF*, *COMT*, *HTR2A*, *MAOA*) have failed to exhibit any associations with depression phenotypes in a much larger sample [5]. Most of the historical candidate genes have also not been supported by GWAS [6]. Two large GWA meta-analyses have been conducted to date. The first one was based on 135,458 cases and 344,901 controls and has identified associations with MDD at 44 independent loci [7]. In the latest study based on 246,363 cases and 561,190 controls, 102 independent variants and 269 genes were found to have a significant association with the disease [8]. Associations were concentrated in genes and gene pathways that are involved in the synaptic organization, neurotransmission, including glutamate neurotransmission (*GRIK5, GRM5*, *GRM8*), and in genes connected to the mechanism of action of antidepressant drugs (e.g., *DRD2*, *GRM5*, *NRG1*).

In recent years, a growing body of evidence highlights a link between glutamate and depression [9,10,11,12]. Ketamine, an antagonist of glutamatergic N-methyl-D-aspartate receptor (NMDAR), demonstrated ultrarapid efficacy in treating refractory depression [13]. L-glutamic acid (glutamate) is the principal excitatory neurotransmitter in the central nervous system [14]. Almost all the glucose that enters the brain is ultimately converted to glutamate [15]. The packaging and release of glutamate are carried out through vesicular glutamate transporters (VGLUTs) [16]. After traversing the synaptic cleft, glutamate binds to cognate ionotropic receptors that are divided into three major classes: N-methyl-D-aspartate (NMDA), α-amino-3-hydroxy-5-methyl-4-isoxazole propionic acid (AMPA) and kainate (KA), as well as to metabotropic glutamate receptors, which have been demonstrated to modulate ion channels activity [17]. The extracellular level of glutamate is strictly regulated by complex machinery. The uptake and redistribution of synaptic glutamate are performed through the membrane excitatory amino acid transporters, EAAT1 and EAAT2, encoded by *SLC1A3* and *SLC1A2,* respectively [18]. SLC1A2 and SLC1A3 transporters are primarily astroglial in distribution [19]. Over 90% of glutamate in the brain is taken up by EAAT2 [20]. A significant reduction in astroglia has been observed in MDD [19,20].

The level of the astrocytic glutamate transporter EAAT2 is decreased in an animal model of depression [21,22]. Furthermore, the blockade of glutamate uptake with EAAT2 inhibitor dihydrokainic acid is sufficient to induce both anhedonia and anxiety in rats [23,24,25]. Hence, an impaired glutamate uptake in the glia could hold an important place in producing specific symptoms of depression. 

In the postmortem study, the gene expression of the membrane transporter SCL1A2 was observed to be diminished in the hippocampal tissue of the MDD group compared with controls [26]. Choudary et al. documented significant downregulation of the glial high-affinity glutamate transporters SLC1A2 in MDD subjects within the anterior cingulate cortex and left dorsolateral prefrontal cortex (DLPFC) [27]. However, in the white matter, *SLC1A2* mRNA was significantly lower in the subjects with MDD compared with the superficial and deep gray matter of DLPFC [28]. A similar downregulation in *SCL1A2* mRNA was also observed in locus coeruleus of patients with the antemortem diagnosis of MDD [29]. Such depletion in glutamate SCL1A2 transporter may result in a significant elevation of extracellular glutamate levels, which is potentially neurotoxic both to neurons and glia [30,31]. In depressed patients, elevated glutamate levels have been reported in the prefrontal and occipital cortex, as well as in the peripheral blood serum [32,33,34]. In addition, an imbalance between neurotransmitters gamma-aminobutyric acid (GABA) and glutamate, resulting in cortical inhibition and excitation, respectively, has been suggested to play a role in MDD [35]. Mallolas et al. detected a novel *SCL1A2* gene promoter polymorphism that was an A-to-C change at 181 bp from the transcriptional start site [36]. Subjects with the mutant genotype, i.e., C allele carriers, were found to have a 30% reduction in promoter activity, which resulted in decreased expression of EAAT2 and, therefore, correlated with greater glutamate concentrations. 

Glycine is a ubiquitous inhibitory neurotransmitter in the central nervous system (CNS), widely expressed in the brainstem and spinal cord [37]. In glutamatergic synapses, glycine acts as an obligatory coagonist of NMDA receptors and modulates excitatory neurotransmission [38]. Two members of the SLC6 family of sodium/chloride-dependent neurotransmitter transporters are responsible for the clearance of synaptic glycine: glycine transporter type 1 (GlyT1) and glycine transporter type 2 (GlyT2) [39]. GlyT1 is strongly associated with glycine uptake in astrocytes and glutamatergic neurotransmission in the cortex and hippocampus, while GlyT2 performs presynaptic glycine uptake on inhibitory glycinergic synapses in the spinal cord, brainstem, and cerebellum [40,41]. In humans, GlyT1 and GlyT2 are encoded by *SLC6A9* and *SLC6A5* genes, respectively [42,43]. Hence, higher synaptic glycine levels could enhance the activation of NMDA receptors. Sarcosine, a GlyT1 inhibitor, was found to produce greater and quicker improvement in depressive symptoms in MDD patients compared with active control of citalopram [44]. 

Considering the findings of the above-cited studies, *SLC1A2, SLC6A5,* and *SLC6A9* have been chosen as the candidate genes in our case-control association study.

In the current work, we focus on examining the association of the *SLC1A2* rs4354668, *SLC6A9* rs2486001, and *SLC6A5* rs2000959 polymorphisms with major depressive disorder and its clinical variables in the Polish population.

## 2. Materials and Methods

### 2.1. Patient and Control Groups Profile

The study began in early 2016, with the first recruitment for the MDD group on the 1st of February and the first control recruited on the 4 February. The last recruitment for MDD and control groups took place on 17 November 2018 and 2 December 2018, respectively. Initially, 293 MDD inpatients meeting the recruitment criteria were invited to participate in the study, of which 132 dropped out at various stages of the study. The MDD group finally consisted of 161 unrelated Caucasian patients matching the diagnostic criteria of DSM-5 (Diagnostic and Statistical Manual of Mental Disorders, Fifth Edition) for major depressive disorder (MDD) at the moment of the recruitment. The DSM-5 criteria were assessed by two clinically trained psychiatrists. The severity of the depression symptoms was evaluated with the Hamilton Depression Rating Scale (HDRS). The bipolar spectrum features were excluded based on the Hirschfeld Mood Disorder Questionnaire and diagnostic criteria of bipolar disorder by Ghaemi [45,46]. The MDD group comprised 116 females (72.1%, mean age 57 ± 12, range 19–84) and 45 males (27.9%, mean age 57 ± 9, range 29–80) recruited from inpatients treated at the Department of Psychiatry and Psychotherapy, the Medical University of Silesia in Katowice, the Neuropsychiatric Hospital in Lubliniec, and the State Hospital for Mental Diseases in Rybnik. Exclusion criteria were assessed by evaluating the patients’ medical records and involved bipolar disorder, bipolar spectrum features, intellectual disability, anxiety disorder, mixed anxiety and depressive disorder, schizoaffective disorder, schizophrenia, organic or substance-related psychosis, autoimmune and chronic inflammatory diseases, and lack of informed consent to participate in the study. All patients were assessed to be capable of giving informed consent to participate in the study. A detailed patients profile is presented in Table 1. The control group included 462 healthy, unrelated individuals, aged from 23 to 68 [females 238 (51.5%), mean age 40 ± 8, range 28–62; males 224 (48.5%), 41 ± 9, range 23–68]. All controls were invited to participate in the study by the research team members in person during blood donations in the Regional Blood Donation and Blood Treatment Center in Katowice. Of the 723 initially invited participants matching the inclusion criteria for the control group, 261 declined. Exclusion criteria for controls were abnormal blood test results, such as elevated inflammatory markers, contagious and autoimmune disease, psychoactive substance abuse except for nicotine, present mental health problems, psychiatric medication, and past mental illness episodes, including family members declared in the questionnaire and based on personal medical examination. Control participants answered personally on-site a standardized questionnaire for blood donors routinely used in Polish Blood Donation Centers. Hirschfeld Mood Disorder Questionnaire was used to rule out participants with bipolar spectrum features. There was a statistically significant difference in age between the MDD and the control group in men (*p* < 0.001) as well as in women (*p* < 0.001). Both males and females were younger in the control group. In the study group, there were more women than men (72.1% vs. 27.9%; *p* < 0.001). We chose not to match controls to patients relying on some latest reports suggesting that matching should be used with great caution, especially in case–control studies, as it may create selection bias and may harm precision and power [47].

All participants were Caucasians of Polish origin living in Upper Silesia. All participants gave their informed consent to participate in the study. The study was approved by the Bioethics Committee of the Medical University of Silesia (resolution KNW/0022/KB1/34/14).

### 2.2. SNP Choice and Genotyping

We selected three SNPs (rs4354668 in the *SLC1A2,* rs2486001 in the *SLC6A9,* and rs2000959 in the *SLC6A5*) with a minor allele frequency (MAF) ≥10% in the European population [48]. Low MAF SNPs are more susceptible to genotyping errors, and they have low power to detect any association for given effect size. Other selection criteria were: assay availability, evidence from the literature (rs4354668) suggesting the potential functional effect of SNP on gene expression [36], and the association of SNPs (rs2486001, rs2000959) with other related diseases, while literature references were strongly taken under consideration [49,50,51,52].

Genomic DNA was extracted from the peripheral blood leucocytes using a QIAamp DNA Blood Mini Kit (Qiagen, Hilden, Germany) according to the manufacturer’s protocol. The purity and concentration of DNA extracts were assessed using a BioPhotometer plus (Eppendorf AG, Hamburg, Germany). The genotyping of SNPs: rs4354668 A/C in the *SLC1A2* and rs2486001 G/A in the *SLC6A9* was performed by PCR-RFLP method as described previously [50], using the following published primers: *SLC1A2* forward 5’-GAGCGGCGGGGCCTCTTTTC-3’ and reverse 5’-TGCAGCCGCTGCCACCTGTG-3’ [51], *SLC6A9* forward 5’-TTCTATTCCCTGGGGTTCAGCA-3’ and reverse 5’-AGCCTGGGCTGAGGCACACCAC-3’ [49]. Amplified products were digested by two restriction enzymes (*Bcn*I—rs4354668 and *Ava*II—rs2486001) (Thermo Fisher Scientific, Lithuania), according to the protocol, and digested products were separated by electrophoresis in 2% agarose gels stained with ethidium bromide. Product sizes were: rs4354668: A allele 381 bp, C allele 262 bp/119 bp; rs2486001: G allele 210 bp, A allele 123 bp/87 bp.

Genotyping of rs2000959 polymorphism in the *SLC6A5* was performed using an allele-specific TaqMan assay (Applied Biosystems, Warrington, UK, catalog number: C__12032096_1_) on a CFX96 Touch™ Real-Time PCR Detection System (Bio-Rad Laboratories, Hercules, CA, USA), in a 96-well format. The PCR reaction was performed in a final volume of 25 μL, containing 10 ng of the DNA template, 12.5 μL TaqMan™ Genotyping Master Mix (Applied Biosystems, Forster City, CA, USA), 1.25 μL of combined primers and probes mix (Applied Biosystems, Warrington, UK), and nuclease-free water. Real-time PCR was performed with a holding stage at 95 °C for 10 min, followed by 40 cycles at 95 °C for 15 s and 60 °C for 1 min. Each analysis was performed with two no-template controls as a negative control, along with three previously genotyped control samples representing particular rs2000959 genotypes as a positive control. Genotyping data were analyzed using Bio-Rad CFX Manager Software 3.1.

As a quality control measure, 5% of randomly selected samples were repeatedly genotyped, and the concordance rate of these repeated samples reached 100%. Samples with missing genotypes have been removed from further analysis.

### 2.3. Statistical Analysis 

Statistical analysis was performed using STATISTICA 13.0 PL (StatSoft, TIBCO Inc., Palo Alto, CA, USA) and R software (R Core Team (2021), R: A language and environment for statistical computing, R Foundation for Statistical Computing, Vienna, Austria, R version 4.0.3 (2020-10-10)) [53]. All tests were two-tailed, and *p* < 0.05 was considered statistically significant. We ensured the power of the test was at an 80% level with the use of the GPower 3.1 software. Nominal and ordinal variables were expressed as percentages, whilst descriptive variables were expressed as mean value ± standard deviation in case of data with normal distribution or as median (lower quartile—upper quartile) in case of data with skewed distribution. Hardy–Weinberg equilibrium (HWE) was examined by Fischer’s exact test to compare the actual genotypes with the expected number. The differences in allele frequencies and genotype distribution between groups were assessed using either the Chi-square (χ2) test or Fisher’s exact test. The distribution of variables was evaluated by the Shapiro–Wilk test and quantile–quantile (Q–Q) plot, and homogeneity of variances was assessed by the Levene test. Logistic regression was applied to calculate the odds ratios (ORs) and 95% confidence intervals (CIs). The logistic regression models (codominant, dominant, recessive, overdominant, and log-additive) were also used to assess potential association with MDD risk, and the best fitting models were determined by the Bayesian information criterion (BIC). The linkage disequilibrium (LD), haplotype analysis, as well as inheritance models (codominant, dominant, recessive, overdominant, and log-additive) were performed using the *SNPAssoc* package in R. The two-way ANOVA (sex, genotype) with Tukey’s post hoc test was used to examine the effect of genotypes on clinical variables (number of episodes, age of onset, duration of the disease, and total HRDS score).

## 3. Results

### 3.1. Comparison of Genotype and Allele Distribution of the Studied Polymorphisms between Patients and Controls

The analysis of the *SLC1A2* rs4354668 polymorphism (Table 2) showed statistically significant differences in the genotype (*p* < 0.001) and allele (*p* < 0.001) distribution between a group of MDD patients and controls. We observed that genotype C/C and allele C were more represented in the MDD group than in the control group. Allele C was statistically significantly associated with MDD [OR = 1.57 (95% CI: 1.25–2.11), *p* < 0.001]. The sex-stratified analysis showed statistically significant differences in the genotype distribution in women (*p* < 0.01). In men, only a trend towards statistical significance was observed (*p* = 0.09). In both female (*p* < 0.01) and male group (*p* < 0.05), statistically significant differences in the allele distribution were found. Moreover, allele C was significantly associated with MDD in both groups: male [OR = 1.66 (95% CI: 1.14–3.04)], female [OR = 1.52 (95% CI: 1.15–2.22)].

The polymorphisms rs2486001 (*SLC6A9*) and rs2000959 (*SLC6A5*) showed no statistically significant difference in the genotype (rs2486001: *p* = 0.53; rs2000959: *p* = 0.71) and allele (rs2486001: *p* = 0.91; rs2000959: *p* = 0.69) distributions between the MDD and the control group (Table 3 and Table 4). The analyses in subgroups stratified according to gender also did not show any statistically significant differences in the genotype (rs2486001: females: *p* = 0.22, males: *p* = 0.97; rs2000959: females: *p* = 0.29, males: *p* = 0.71) and allele distributions (rs2486001: females: *p* = 0.55, males: *p* = 0.94; rs2000959: females: *p* = 0.26, males: *p* = 0.40).

None of the three analyzed SNPs did not depart from the Hardy–Weinberg equilibrium both in the study [rs4354668 (*p* = 0.16), rs2486001 (*p* = 0.53), rs2000959 (*p* = 0.48)] and in the control group [rs4354668 (*p* = 0.39), rs2486001 (*p* = 0.60), rs2000959 (*p* > 0.99)].

In the next step, we explored the potential association between MDD and individual polymorphisms on the basis of different genetic inheritance models (dominant, recessive, codominant, and overdominant). Genotypes of the two analyzed polymorphisms (rs2486001, rs2000959) were not significantly associated with MDD in any of the inheritance models.

The *SCL1A2* SNP genotypes were significantly associated with MDD in the codominant and recessive models. Genotype C/C was significantly associated with MDD in both models [codominant: OR = 2.50 (95% CI: 1.52–4.17, *p* < 0.001), recessive: OR = 2.38 (95% CI: 1.56–3.70, *p* < 0.001)]. After gender stratification, significant differences were observed only in females [codominant: OR = 2.50 (95% CI: 1.33–4.76, *p* < 0.01), recessive: OR = 2.56 (95% CI: 1.52–4.35, *p* < 0.001)]. In males, the differences were not statistically significant; however, in the recessive model, a trend towards statistical significance was observed (*p* = 0.052) (Table 5). 

To evaluate the combined association effect of the three polymorphisms, we performed a haplotype analysis. This haplotype analysis revealed statistically significant differences in the frequency of haplotype CGC between the MDD and the control group. The patients with CGC haplotype are 1.75-times more likely to have MDD than those with the most frequent haplotype AGC [OR = 1.75 (95% CI: 1.16–2.63, *p* < 0.01)]. Such differences were not observed after stratification according to gender; however, males with CGC haplotype showed a trend towards statistical significance (*p* = 0.06) (Table 6).

### 3.2. Correlation between Gender, Genotype, and Clinical Variables of MDD

Two-way ANOVA showed no statistically significant association of rs4354668, rs2486001, and rs2000959 genotypes on clinical variables of MDD (Table 7). However, for rs2486001 SNP within *SLC6A9*, a tendency towards a statistically significant effect of the interaction gender x genotype on mean HDRS score was observed (*p* = 0.08). Females with rs2486001 G/G genotype had higher mean HDRS score than those with G/A genotype (15.2 ± 9.1 vs. 11.1 ± 6.8), whereas males with rs2486001 G/A genotype had higher mean HDRS score than those with G/G genotype (18.4 ± 8.2 vs. 16.2 ± 10.3). Patients with rs2486001genotype A/A were excluded from the analysis because of the scarce sample number. Moreover, a statistically significant association between gender and the duration of the disease was detected for each of the studied SNP [A/C genotype of rs4354668: F vs. M median duration of the disease 11 vs. 6 years (*p* < 0.01); G/G genotype of rs2486001: F vs. M median duration of the disease 11 vs. 8 years (*p* < 0.01); C/A genotype of rs2000959: F vs. M median duration of the disease 12 vs. 6 years (*p* < 0.01)].

## 4. Discussion

The results of the present study suggest an association of rs4354668 SLC1A2 polymorphism with the presence of MDD in the Polish sample of participants with MDD and controls. We found that the genotype C/C and C allele were more frequent in depressed patients than in healthy ones. According to our understanding, C allele carriers have 30% lower promoter activity, which results in decreased expression of EAAT2 and, therefore, correlates with greater glutamate concentrations that are found to be potentially neurotoxic to neurons. As far as we know, only a few studies on the genetic polymorphism of *SLC1A2* in MDD were conducted. The majority of studies reported an association of *SLC1A2* SNPs with bipolar disorder [54,55,56,57] and schizophrenia [58,59]. Polymorphism rs4354668 evaluated in this paper was also analyzed in a Thai sample [60]. In contrast to our findings, no significantly different genotype/allele distributions of *SLC1A2* rs4354668 were found between patients in the MDD group and volunteers in the control group. Various problems may reasonably lead to such discrepancies between the studies. A possible heterogeneity of used diagnostic methods and applied clinical criteria of inclusion and exclusion may be one of them. Bipolar spectrum features are often underdiagnosed, and it is believed that about one-third of MDD diagnoses might belong to the bipolar spectrum [61,62]. Therefore, we aimed to increase the accuracy of diagnosis by using a slightly stricter protocol that, in contrast to the above-cited study, encompassed ruling out symptoms of bipolar disorder and bipolar spectrum features based on standardized questionnaires. Both in our study and the study conducted in Thailand, no data on family history of mental disorders were collected, so one population might have been more heavily genetically predisposed to develop MDD than the other one. Moreover, different gene–gene and gene–environment interactions across various populations may also contribute to the inconsistency of the results between studies. Our study population comprised a significantly higher number of participants (161 vs. 100) with a similar female-to-male ratio that might have also impacted the statistical analysis. Interestingly, the distribution of genotypes in Thai healthy controls (*n* = 100) differs greatly from the Caucasian population analyzed in our study (*n* = 462): 7 participants had the T/T genotype (7%), 47 participants had the G/T genotype (47%), and 46 participants had the G/G genotype (46%). In the current paper, 154 participants (33.33%) in the control group had an A/A genotype, 234 participants (50.65%) had an A/C genotype, and 74 participants (16.02%) had a C/C genotype. Both populations were in Hardy–Weinberg equilibrium. Therefore, the genetic differences among ethnic groups could result in the predominance of A/A homozygotes and the paucity of C/C homozygotes in the case of our study. Further research is needed to establish the role of rs4354668 polymorphism in MDD development across different ethnic groups. So far, no other studies of *SLC1A2* rs4354668 polymorphism in MDD have been reported. 

Interesting results in our study were sex differences in the rs4354668 genotype distributions. Sex-stratified analysis showed that the genotypes of rs4354668 were significantly associated with MDD only in women based on the results of the codominant (OR = 2.50, *p* < 0.01) and recessive models (OR = 2.56, *p* < 0.001). These findings suggest that in the case of rs4354668 polymorphism, sex may modulate the association with MDD. However, it cannot be excluded that the observed sex-specific associations may result from a small group of the studied men, especially as in the recessive model, a strong tendency towards statistical significance was observed in men (*p* = 0.052). In turn, statistically significant differences in allele frequencies were observed in both men and women subgroups. On the other hand, previous studies have reported female-specific variants for depressive disorder, and a more recent study provides genetic evidence that the increased prevalence of depressive disorders in women can be attributed to inherited variants [63,64]. Estrogen has well-known protective actions in the CNS, achieved through the regulation of synaptic function, synaptic plasticity, and neurogenesis [65]. Furthermore, growing evidence suggests that estrogen may mediate its neuroprotection via excitatory neurotransmitter systems [66]. For example, estrogen has been shown to enhance the expression of the EAAT2 (GLT-1) glutamate transporter, which increases glutamate uptake by astrocytes and protects neurons from glutamate toxicity [67]. Despite the many potential benefits of estrogen, women with the SLC1A2 rs4354668 C/C genotype were found to be at risk for developing MDD, and similar associations were not observed in men. Since EAAT2 is a critical astrocytic glutamate transporter that maintains extracellular glutamate below excitotoxic levels, we speculate that the presence of the SLC1A2 rs4354668 C/C genotype (associated with a lower expression of EAAT2) may partially attenuate estrogen-mediated neuroprotection and lead to an increased susceptibility to MDD. However, further analyses are needed to elucidate the role of the SLC1A2 rs4354668 variant in the development of MDD in women.

A particular gene-by-environment interaction of early experienced stress and the functional polymorphism rs4354668 in *SLC1A2* has been shown in the hippocampal gray matter in subjects with bipolar disorder [55]. After exposure to higher levels of adverse childhood experiences, G/G homozygotes were found to be more vulnerable to stress, reporting lower gray matter volume in the hippocampus. As the mutant G allele is associated with less transporter expression and a 30% reduction in promoter activity compared with the T allele, the levels of synaptic glutamate have been hypothesized to be increased and could lead to neurotoxicity and brain damage [31]. Morphometric studies of individuals with bipolar disorder showed reduced gray matter volumes in the hippocampal formation, unlike individuals with unipolar depression, who showed reduced gray matter volumes in the anterior cingulate gyrus [68]. Two different SNPs (rs3812778 and rs3829280) of the *SLC1A2* were associated with the level of glutamate within the anterior cingulate in both bipolar and unipolar depression, with minor allele carriers having significantly higher glutamate levels in comparison with common allele homozygotes [69]. In the study of Zhang et al., the rs4354668 SNP in the *SLC1A2* gene was significantly associated with schizophrenia, and there were better executive function performances in all subjects homozygous for the T allele compared with the G allele carriers [59].These findings are consistent with previous reports demonstrating that the rs4354668 G allele has a disadvantageous effect on cognitive functions such as working memory and executive functions [70]. Cognitive impairment is believed to be a core feature of depression [71]. Therefore, the lack of assessment of cognitive deficits seems to be an important limitation in the present study. Further work is encouraged to delineate the impact of polymorphisms in the glutamate system on cognitive performances in a depressed population. 

Lithium salts are a common mood stabilizer, highly effective in the treatment of the bipolar disorder. The *SLC1A2* rs4354668 polymorphism was reported to influence the total episode recurrence rate and the efficacy of lithium treatment response in a sample of Italian patients with bipolar disorder [54]. These findings seem to be particularly interesting in the context of lithium in the augmentation strategies of antidepressant medications. The World Federation of Societies of Biological Psychiatry Task Force recommends lithium as the first-line augmentation option for treatment-resistant depression, with a recent meta-analysis supporting this approach [72,73].

We did not analyze the potential influence of the studied SNPs on suicidal behaviors among depressed patients. Since suicide is inseparably linked to mood disorders, that is a serious limitation of this study. No research on the association of *SLC1A2* rs4354668 polymorphism with suicidal behavior has been conducted to date. However, in the Irish study of Murphy et al., another *SLC1A2* variant, rs4755404, was associated with suicide attempts, but the findings are inconsistent with other researchers’ results [74]. Hee-Yeon Choi et al. studied various psychological and genetic risk factors for suicidal behaviors in the Korean population and did not find any significant associations between the rs4755404 *SCL1A2* variant and suicidal behavior [75]. Similarly, the Italian researchers failed to replicate these results [57]. Thus, the role of *SLC1A2* polymorphisms in suicide remains to be clarified.

The results of our study indicate no association of SNPs in genes involved in the glycinergic neurotransmission, i.e., rs2486001 of *SLC6A9* and rs2000959 of *SLC6A5* with the occurrence of MDD. To our knowledge, the associations of these two SNPs with MDD have not been reported previously. The number of published association studies regarding glycine transporters is scarce and mostly limited to schizophrenia and substance use disorder. Tsai et al. observed no significant differences in allelic frequencies or genotypic distributions of the *SCL6A9* polymorphisms (rs1766967, rs16831541, rs2248632, and rs2248253) between the group of patients with schizophrenia and healthy controls in the Chinese population [76]. In addition, Deng et al., who tested various SNPs in glycine transporters genes, concluded that *SLC6A9* is unlikely to be a major susceptibility gene for schizophrenia in the Japanese population [51]. Morita et al. examined three SNPs of the *SLC6A9* (rs2486001, rs2248829, rs2248632) in the Japanese sample of methamphetamine-addicted subjects and found that the TG haplotype consisting of rs2486001 and rs2248829 SNPs approximately doubled the risk of predisposition to methamphetamine-use disorder [49]. The rs2486001 polymorphism in *SLC6A9* and four other SNPs in *SLC6A9* and *SLC6A5* (currently not analyzed) was not associated with alcohol dependence in the group of 644 German alcohol-dependent subjects [77]. Noteworthy, the interest in glycine transporters has been continuously growing in recent years and encompasses numerous fields of medicine. Mutations in *SLC6A9* have been linked to the development of essential hypertension and have become causal factors of glycine encephalopathy [78,79].

Several limitations of our study must be considered. Firstly, the sample size of the MDD group was relatively small compared with other studies in this field, which might contribute to false positive or false negative results; thus, replication studies with larger sample sizes and samples from different ethnicities are necessary. Secondly, the MDD group comprised a significantly higher number of females than males. Future studies should include assessments of cognitive function and suicidal attempts to determine how these variables could influence/confound the association between these polymorphisms and MDD. Thirdly, *SLC1A2*, *SLC6A9,* and *SLC6A5* genes are outside the GWAS-indexed risk loci for MDD. Moreover, the significantly younger mean age of the control group in comparison with the MDD group is a serious limitation to our study since many of these control subjects may still develop MDD in the future. Furthermore, the gender distribution is not similar among groups with more MDD female patients in the MDD group, which may impact the obtained results. Thus, further studies with more balanced groups in age and gender are needed to confirm our findings. Finally, we analyzed only one SNP of each studied gene. Therefore, it cannot be excluded that other SNPs of the *SLC1A2*, *SLC6A9,* and *SLC6A* might be associated with MDD or its clinical features. 

Overall, we conducted our study on a relatively significant number of patients with depression of Caucasian ethnicity in the presence of the control group. Bearing in mind all the limitations of the current paper, we cautiously speculate that the mutant *SLC1A2* carriers are more susceptible to developing MDD. Further studies with more SNPs and larger sample size are needed to establish the role of *SLC1A2*, *SLC6A9*, and *SLC6A* polymorphisms in the pathogenesis of depressive disorders.

## 5. Conclusions

In conclusion, the current study indicates an association of rs4354668 polymorphism in *SCL1A2* with the occurrence of MDD in a Polish sample. The frequency of genotype C/C and allele C was twice as high in the MDD group as in the control group. The analysis of different genetic inheritance models showed that the genotype C/C of the rs4354668 polymorphism was significantly associated with MDD in the codominant and recessive models. Haplotype analysis revealed that patients with CGC haplotype are 1.75-times more likely to have depression than those with the most frequent haplotype AGC. No associations between *SLC6A9* and *SLC6A5* polymorphisms and susceptibility to MDD were observed. Up to now, this is the first study of SLC1A2, SLC6A5, and SLC6A9 polymorphisms in the Polish MDD population; thus, we believe it may contribute to delineating new areas of interest for further Polish genetic studies.

## Figures and Tables

**Table 1 jcm-11-05914-t001:** Detailed patients profile.

Variable	Total Group			Females			Males		
	Mean (±SD)	Median (Q_1_–Q_3_)	Range	Mean (±SD)	Median (Q_1_–Q_3_)	Range	Mean (±SD)	Median (Q_1_–Q_3_)	Range
Number of episodes	5.6 ± 6	4 (3–6)	1–50	6 ± 5	4 (3–7)	2–35	6 ± 7	4 (3–5)	1–50
Duration of the disease (years)	12 ± 10	10 (5–17)	0.5–47	13 ± 10	10 (6–18)	1–47	10 ± 8	7 (3–15)	0.5–30
Total HDRS score	14.8 ± 9	13 (7–21)	0–42	14.1 ± 8.7	13 (7–20)	0–40	16.5 ± 9.8	15 (9–23)	2–42
Age of onset	45 ± 12	45 (36–53)	14–70	44 ± 12	43 (35–52)	14–70	47 ± 12	48 (40–55)	21–68

SD—standard deviation; Q_1_—lower quartile; Q_3_—upper quartile.

**Table 2 jcm-11-05914-t002:** Analysis of genotype and allele distributions in the entire population; stratified by gender (codominant inheritance model) of *SLC1A2* rs4354668 polymorphism.

SNP		Total Group		χ^2^	*p*-Value	Females		χ^2^	*p*-Value	Males		χ^2^	*p*-Value
		Patients	Controls			Patients	Controls			Patients	Controls		
**Genotypes**													
**rs4354668**	A/A	41 (25.47%)	154 (33.33%)	15.98	**<0.001**	30 (25.86%)	73 (30.67%)	12.58	**<0.01**	11 (24.44%)	81 (36.16%)	4.80	**0.09**
	A/C	71 (44.10%)	234 (50.65%)			51 (43.97%)	131 (55.04%)			20 (44.45%)	103 (45.98%)		
	C/C	49 (30.43%)	74 (16.02%)			35 (30.17%)	34 (14.29%)			14 (31.11%)	40 (17.86%)		
**Alleles**													
**rs4354668**	A	153 (47.52%)	542 (58.66%)	12.02	**<0.001**	111 (47.84%)	277 (58.19%)	6.74	**<0.01**	42 (46.67%)	265 (59.15%)	4.77	**<0.05**
	C	169 (52.48%)	382 (41.34%)			121 (52.16%)	199 (41.81%)			48 (53.33%)	183 (40.85%)		

Nominal associations and trends towards statistical significance are bolded.

**Table 3 jcm-11-05914-t003:** Analysis of genotype and allele distributions in the entire population; stratified by gender (codominant inheritance model) of *SLC6A9* rs2486001 polymorphism.

SNP		Total Group		χ^2^	*p*-Value	Females		χ^2^	*p*-Value	Males		χ^2^	*p*-Value
		Patients	Controls			Patients	Controls			Patients	Controls		
**Genotypes**													
**rs2486001**	A/A	2 (1.24%)	13 (2.81%)	1.26	0.53	1 (0.86%)	9 (3.78%)	3.05	0.22	1 (2.22%)	4 (1.79%)	0.71	0.97
	A/C	43 (26.71%)	121 (26.19%)			31 (26.72%)	58 (24.37%)			12 (26.67%)	63 (28.13%)		
	C/C	116 (72.05%)	328 (71.00%)			84 (72.41%)	171 (71.85%)			32 (71.11%)	157 (70.09%)		
**Alleles**													
**rs2486001**	A	51 (4.08%)	147 (11.76%)	0.01	0.91	33 (12.28%)	76 (43.50%)	0.36	0.55	14 (7.81%)	71 (49.26%)	0.00	0.94
	C	275 (26.08%)	777 (73.92%)			199 (13.56%)	400 (30.66%)			76 (8.92%)	377 (34.01%)		

**Table 4 jcm-11-05914-t004:** Analysis of genotype and allele distributions in the entire population; stratified by gender (codominant inheritance model) of *SLC6A5* rs2000959 polymorphism.

SNP		Total Group		χ^2^	*p*-Value	Females		χ^2^	*p*-Value	Males		χ^2^	*p*-Value
		Patients	Controls			Patients	Controls			Patients	Controls		
**Genotypes**													
**rs2000959**	A/A	20 (12.42%)	47 (10.17%)	0.69	0.71	16 (13.79%)	20 (8.40%)	2.48	0.29	4 (8.89%)	27 (12.05%)	0.69	0.71
	A/C	67 (41.46%)	202 (43.72%)			49 (42.24%)	106 (44.54%)			18 (40.00%)	96 (42.86%)		
	C/C	74 (45.96%)	213 (46.10%			51 (43.97%)	112 (47.06%)			23 (51.11%)	101 (45.09%)		
**Alleles**													
**rs2000959**	A	107 (8.59%)	296 (23.76%)	0.16	0.69	81 (12.28%)	146 (43.50%)	0.72	0.40	26 (4.83%)	150 (27.88%)	0.72	0.40
	C	215 (17.26%)	628 (50.40%)			151 (13.56%)	330 (30.66%)			64 (11.90%)	298 (55.39%)		

**Table 5 jcm-11-05914-t005:** Analysis of different inheritance models for the SNP rs4354668 in *SLC1A2* between MDD and control group.

Model	Genotype	Total Group (*n* = 623)					Females (*n* = 354)					Males (*n* = 269)				
		Depression	Control	OR (95% CI)	*p*- Value	BIC	Depression	Control	OR (95% CI)	*p*-Value	BIC	Depression	Control	OR (95% CI)	*p*-Value	BIC
**Codominant**	A/A	41 (25.5%)	154 (33.3%)	1.00	**<0.001**	700.9	30 (25.9%)	73 (30.7%)	1.00	**<0.01**	453.4	11 (24.4%)	81 (36.2%)	1.00	0.10	255.2
	A/C	71 (44.1%)	234 (50.6%)	1.08 (0.69–1.67)			51 (44%)	131 (55%)	0.94 (0.56–1.61)			20 (44.4%)	103 (46%)	1.43 (0.65–3.13)		
	C/C	49 (30.4%)	74 (16.0%)	**2.50 (1.52–4.17)**			35 (30.2%)	34 (14.3%)	**2.50 (1.33–4.76)**			14 (31.1%)	40 (17.9%)	2.56 (1.08–6.25)		
**Dominant**	A/A	41 (25.2%)	154 (33.3%)	1.00	0.10	707.3	30 (25.9%)	73 (30.7%)	1.00	0.35	458.7	11 (24.4%)	81 (36.2%)	1.00	0.12	251.7
	A/C-C/C	120 (74.5%)	308 (66.7%)	1.41 (0.93–2.13)			86 (74.1%)	165 (69.3%)	1.27 (0.77–2.08)			34 (75.6%)	143 (63.8%)	1.75 (0.84–3.70)		
**Recessive**	A/A-A/C	112 (69.6%)	388 (84.0%)	1.00	**<0.001**	694.6	81 (69.8%)	204 (85.7%)	1.00	**<0.001**	447.6	31 (68.9%)	184 (82.1%)	1.00	0.052	250.4
	C/C	49 (30.4%)	74 (16%)	**2.38 (1.56–3.70)**			35 (30.2%)	34 (14.3%)	**2.56 (1.52–4.35)**			14 (31.1%)	40 (17.9%)	2.08 (1.01–4.35)		
**Overdominant**	A/A-C/C	90 (55.9%)	228 (49.4%)	1.00	0.09	707.1	65 (56%)	107 (45%)	1.00	**0.05**	455.7	25 (55.6%)	121 (54%)	1.00	0.85	254.1
	A/C	71 (44.1%)	234 (50.6%)	0.72 (0.50–1.05)			51 (44%)	131 (55%)	0.64 (0.41–1.00)			20 (44.4%)	103 (46%)	0.94 (1.49–1.79)		

OR—odds ratio; CI—confidence interval; BIC—Bayesian information criterion. Nominal associations are bolded. The linkage disequilibrium analysis showed a weak linkage disequilibrium (LD) only between the polymorphisms rs2486001 and rs2000959 (D’ = 0.097, r = 0.060, *p* < 0.05). The polymorphisms rs4354668 and rs2000959 showed only a trend to LD (D’ = 0.071, r = 0.055, *p* = 0.053).

**Table 6 jcm-11-05914-t006:** Haplotype analysis of the studied polymorphisms in patients with MDD and control subjects.

Haplotype			Total Group (*n* = 623)			Females (*n* = 354)			Males (*n* = 269)		
rs4354668	rs2486001	rs2000959	Freq (%)	OR (95% CI)	*p*-Value	Freq (%)	OR (95% CI)	*p*-Value	Freq (%)	OR (95% CI)	*p*-Value
A	G	C	33.89	1.00	---	31.29	1.00	---	36.65	1.00	---
C	G	C	24.27	1.75 (1.16–2.63)	**<0.01**	26.34	0.63 (0.37–1.05)	0.12	21.97	0.53 (0.27–1.04)	**0.06**
A	G	A	13.68	1.23 (0.74–2.08)	0.41	15.19	0.66 (0.36–1.21)	0.30	12.44	1.73 (0.52–5.70)	0.41
C	G	A	12.59	1.28 (0.80–2.08)	0.30	11.78	0.75 (0.41–1.35)	0.29	13.14	0.83 (0.36–1.89)	0.65
A	A	C	5.32	0.52 (0.18–1.47)	0.22	6.30	3.06 (0.63–14.76)	0.13	4.28	1.18 (0.27–5.13)	0.95
C	A	C	4.18	1.54 (0.67–3.57)	0.31	4.01	0.44 (0.15–1.24)	0.10	4.39	1.19 (0.24–5.86)	0.80
C	A	A	3.18	2.04 (0.77–5.56)	0.15	3.07	0.34 (0.09–1.26)	0.19	3.44	0.75 (0.15–3.77)	0.67
A	A	A	2.89	1.32 (0.41–4.35)	0.64	2.02	1.05 (0.18–6.00)	0.76	3.70	0.49 (0.10–2.31)	0.47
**Global haplotype association *p*-value <0.05**											

OR—odds ratio; CI—confidence interval. Nominal associations and trends towards statistical significance are bolded.

**Table 7 jcm-11-05914-t007:** Results from the two-way ANOVA on clinical variables of MDD.

Variable	rs4354668			rs2486001			rs2000959		
	Sex	Genotype	Sex × Genotype	Sex	Genotype ^1^	Sex × Genotype	Sex	Genotype	Sex × Genotype
Age of onset	0.10	0.27	0.33	0.57	0.23	0.85	0.33	0.46	0.82
Number of episodes	0.73	0.24	0.20	0.71	0.73	0.31	0.79	0.68	0.89
Duration of the disease	**<0.01**	0.40	0.64	**<0.05**	0.91	0.51	**<0.01**	0.45	0.19
Total HDRS score	0.15	0.28	0.15	**<0.05**	0.60	**0.08**	0.45	0.45	0.69

^1^A/A genotype was excluded because of the scarce sample number. Nominally significant *p*-values and trends towards statistical significance are bolded.

## Data Availability

The datasets generated and analyzed during the current study are not publicly available but are available from the corresponding author upon reasonable request.

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
