# Peer review of "Association Study of the SLC1A2 (rs4354668), SLC6A9 (rs2486001), and SLC6A5 (rs2000959) Polymorphisms in Major Depressive Disorder"

_jcm, 2022, doi:10.3390/jcm11195914_

Round 1

Reviewer 1 Report (Previous Reviewer 3)

Dear Editor,

Thank you for your invitation to review “Association study of the SLC1A2 (rs4354668), SLC6A9 (rs2486001) and SLC6A5 (rs2000959) polymorphisms in Major Depressive Disorder" for your journal. 

This paper aimed to determine whether polymorphisms of SCL1A2, SCL6A5 and SCL6A9 may associate with Major Depressive Disorder (MDD), by investigating a sample of 161 Caucasian MDD patients and 462 healthy controls.

Overall, the study is interesting, however there are a number of shortcomings, especially in the Methods’ and Results sections, that should be taken into consideration.

Comments for the authors

The study entitled “Association study of the SLC1A2 (rs4354668), SLC6A9 (rs2486001) and SLC6A5 (rs2000959) polymorphisms in Major Depressive Disorder” addresses an important topic with regard to factors that are likely to associate with an increased risk of Major Depressive Disorder (MDD); in particular the study aimed to determine whether polymorphisms of SCL1A2, SCL6A5 and SCL6A9 may associate with MDD, by investigating a sample of 161 Caucasian MDD patients and 462 healthy controls.

Overall, the study is interesting, however some specific areas to attend are noted below.

Methods

  1. The authors state in the limitations (line 377) that as compared to other relevant studies, their sample size was relatively low. However, in line 188 of the Methods’ section, it is mentioned that an a priori statistical power analysis to determine sample size was used and adequate power was assured. Given the low sample size, this finding is rather important for the study. It could be very useful if the authors provided more information regarding the power analysis procedure and even include a detailed description in a supplementary file.
  2.  Furthermore, regarding the sample size, it is important that the authors were able to include a larger number of controls, as it is supported that a higher controls’ sample size (up to a ratio 1:5) may increase the overall power.

However, it would be preferable - for statistical analysis’ reasons - if these controls were individually matched to the patients. Could the authors provide information for the reasons that the controls were not matched to patients, especially regarding important demographic characteristics, such as sex and age? How were the controls selected? While, the authors provide information (lines 133- 140) it is suggested that a more detailed description regarding their sampling procedure is included.

  1. Regarding the exclusion criteria of the controls, the authors state that these included “psychoactive substance abuse except for nicotine, past mental illness episodes including family members declared in the questionnaire”. It appears, that information regarding mental health history etc. were based only on self-report questions and not on a more objective measure (eg a structured clinical interview by a certified clinician). Therefore, information bias may have occurred as a result of respondent misunderstanding, while controls with a mental health problem that was not diagnosed by a mental health professional at the time, may have wrongfully been included in the study. The authors should provide more information regarding the above in the Methods’ section or include this as a limitation.
  2. Lines 133-134: Please check and correct the controls’ age range (i.e total controls sample is 18-65 years of age, of females 28-62 and of the males 23-68)

Results

1.     Line 213: It is mentioned that “the allele C statistically significantly increased the risk of developing MDD (OR=1.57.etc)”. However, words that may imply a causal relationship (i.e. developing MDD) should be avoided, given that this is a cross-sectional study and the patients were assessed after being diagnosed, that is after having developed MDD; finding a relationship does not necessarily mean a causal relationship exists. (In other parts of the paper such words/phrases are used as well, eg line 218 etc)

2.     Please include exact p-values instead of <0.01 or <0.05 (eg Table 2 and line 216 etc.)

3.     It is recommended that the results presented in lines 220-229, be included in a table (if not possible in the main manuscripts as a supplement file) as in their current form only p-values are provided, without any descriptive statistics or confidence intervals to offer a better overall understanding for the readers.

4.     Line 229: Please change p=1.00 to p>0.99

5.     It is strongly recommended that Table 5 is modified in order to include descriptive statistics (mean, sd) of the variables and effect sizes (F/df of main effects and interactions).

Discussion

1.     In lines 347-348, the authors state that a serious limitation of their study is that they did not analyse the influence of the studied SNPs on suicidal behaviors. In spite of the limitation, it would be useful if the authors could provide descriptive data (frequency and relative frequency) of MDD patients in the sample that had demonstrated suicidal behavior.

Author Response

Reviewer 2 Report (New Reviewer)

  1. Provide rs numbers in the abstract.
  1. Major problem is a big difference of average age between case and control groups. There are over 17 years between two groups. It may contain confounding factors.
  1. The reported associations of specific alleles with depression risk and sex are interesting but lack details of how these associations interacted with sex. It would help if authors present figures of each of these results in addition to tables and present statistics of the possible sex-specific interactions both for the risk  associations.

4. The limited sample sizes are potentially limiting, and it may help if authors explicitly acknowledge the sample sizes as a limitation, in addition to suggesting the best next steps.

5. The authors should add more on the importance of their findings as they relate to the importance of  genetic studies in the Polish population.  

6. The authors reported a positive association, however, it not is clear how this gene or polymorphisms can participate in the development to the disease. Add a proposed model or theoretical diagram suggesting the possible role that this would play in the development of the disease based on their results.

Author Response

Please see the attachment.Please see the attachment.

Round 2

Reviewer 1 Report (Previous Reviewer 3)

The authors have answered to most comments sufficiently.

 However, regarding comments on the Results section (#3 and #5) it is suggested that authors make the respective changes in the manuscript.

Reporting the p value alone may be misleading and should be avoided. It is preferable to report the mean values (SDs) for each group, the 95% confidence interval, and then the p value.

 In particular:

·         It is recommended that the results presented in lines 220-229, be included in a table (if not possible in the main manuscripts as a supplement file) as in their current form only p-values are provided, without any descriptive statistics or confidence intervals to offer a better overall understanding for the readers.

·         It is strongly recommended that Table 5 is modified in order to include descriptive statistics (mean, sd) of the variables and effect sizes (F/df of main effects and interactions).

Author Response

We have provided two additional tables with lacking descriptive statistics according to reviewer's suggestion. 

This manuscript is a resubmission of an earlier submission. The following is a list of the peer review reports and author responses from that submission.

Round 1

Reviewer 1 Report

Rodek and colleagues present an investigation of three candidate gene polymorphisms for MDD. The paper is well written and clearly presented, but suffers from major limitations:

Both the research hypothesis and the discussion of results are not grounded in our current understand of the genetics of highly polygenic complex traits such as major depression. Numerous genome wide association studies with samples multiple orders of magnitude larger than that of the current studies have taught us that common variants have small effects. An odds ratio of 1.57 (that reported here for rs2486001) would make it far and away the largest genetic contributor to depression; coming up with a measurement error model or theory of population-specificity that could explain why multiple large GWAS in European and Asian populations (eg., Wray, 2018; CONVERGE consortium, 2015) failed to implicate it would be challenging. However, the authors don't even mention the genome-wide literature at all, nor do they address the widely discussed limitations of the candidate gene approach: historic candidate genes for MDD have failed to replicate (e.g. Border, 2019), and the general failure of the candidate gene approach has led the National Institute of Mental Health to stop funding this approach entirely.

Reviewer 2 Report

This manuscript describes a candidate gene study of three SNPs (rs4354668 in SLC1A2, rs2486001 in SLC6A9 and rs2000959 in SLC6A5) and their association with major depressive disorder (MDD) in a sample of 161 cases and 462 controls. The authors report an association of the rs4354668 genotype and allelic distribution with MDD diagnosis.

There are a number of concerns I have with this manuscript, listed below:

  1. It is now recognized in the field of psychiatric genetics that candidate gene studies are an inadequate design for the study of genetic associations with complex psychiatric phenotypes, such as MDD (see report from NIMH: https://www.nimh.nih.gov/about/advisory-boards-and-groups/namhc/reports/report-of-the-national-advisory-mental-health-council-workgroup-on-genomics). This is because there are many genetic variants across the genome that influence risk for MDD, and each of those variants has a very small effect size; because of the small effect sizes of those variants, very large sample sizes are needed to have adequate statistical power to detect genetic associations at SNPs. Furthermore, because of the genome-wide distribution of risk variants, scientists have found that genome-wide association studies, which take a “hypothesis-free” approach rather than selecting specific SNPS, are a better approach for studying genetic associations with MDD. Please see Border et al. (2019) for a comparison of candidate gene vs. GWAS findings for MDD: https://ajp.psychiatryonline.org/doi/10.1176/appi.ajp.2018.18070881 and Duncan et al. (2019) for a helpful discussion: https://www.ncbi.nlm.nih.gov/pmc/articles/PMC6785091/
  2. Related to the concern above, the studies that the authors cite as their rationale for studying these three SNPS are all candidate gene studies themselves [references 49, 50 and 51]; none of the evidence for these SNPs comes from an unbiased, hypothesis-free, genome-wide study (unless I missed it).
  3. The authors don’t seem to account for or control for genetic ancestry in their analyses (e.g., by including genetic ancestry principal components as covariates in their analysis). This is essential, even if their study group is relatively “homogeneous” in self-reported race and ethnicity.
  4. Finally, the authors don’t seem to do any corrections for multiple testing, despite performing a number of analyses (three primary analyses, sex-stratified analyses for each SNP, haplotype analyses, gender x genotype interactions, etc).

Reviewer 3 Report

The study entitled “Association study of the SLC1A2 (rs4354668), SLC6A9 (rs2486001) and SLC6A5 (rs2000959) polymorphisms in Major Depressive Disorder” addresses an important topic with regard to factors that are likely to associate with an increased risk of Major Depressive Disorder (MDD); in particular the study aimed to determine whether polymorphisms of SCL1A2, SCL6A5 and SCL6A9 may associate with MDD, by investigating a sample of 161 Caucasian MDD patients and 462 healthy controls.

Overall, the study is interesting, however some specific areas to attend are noted below.

Methods

  1. The authors state in the limitations (line 377) that as compared to other relevant studies, their sample size was relatively low. However, in line 188 of the Methods’ section, it is mentioned that an a priori statistical power analysis to determine sample size was used and adequate power was assured. Given the low sample size, this finding is rather important for the study. It could be very useful if the authors provided more information regarding the power analysis procedure and even include a detailed description in a supplementary file.
  2.  Furthermore, regarding the sample size, it is important that the authors were able to include a larger number of controls, as it is supported that a higher controls’ sample size (up to a ratio 1:5) may increase the overall power.

However, it would be preferable - for statistical analysis’ reasons - if these controls were individually matched to the patients. Could the authors provide information for the reasons that the controls were not matched to patients, especially regarding important demographic characteristics, such as sex and age? How were the controls selected? While, the authors provide information (lines 133- 140) it is suggested that a more detailed description regarding their sampling procedure is included.

  1. Regarding the exclusion criteria of the controls, the authors state that these included “psychoactive substance abuse except for nicotine, past mental illness episodes including family members declared in the questionnaire”. It appears, that information regarding mental health history etc. were based only on self-report questions and not on a more objective measure (eg a structured clinical interview by a certified clinician). Therefore, information bias may have occurred as a result of respondent misunderstanding, while controls with a mental health problem that was not diagnosed by a mental health professional at the time, may have wrongfully been included in the study. The authors should provide more information regarding the above in the Methods’ section or include this as a limitation.
  2. Lines 133-134: Please check and correct the controls’ age range (i.e total controls sample is 18-65 years of age, of females 28-62 and of the males 23-68)

Results

  1. Line 213: It is mentioned that “the allele C statistically significantly increased the risk of developing MDD (OR=1.57.etc)”. However, words that may imply a causal relationship (i.e. developing MDD) should be avoided, given that this is a cross-sectional study and the patients were assessed after being diagnosed, that is after having developed MDD; finding a relationship does not necessarily mean a causal relationship exists. (In other parts of the paper such words/phrases are used as well, eg line 218 etc)
  2. Please include exact p-values instead of <0.01 or <0.05 (eg Table 2 and line 216 etc.)
  3. It is recommended that the results presented in lines 220-229, be included in a table (if not possible in the main manuscripts as a supplement file) as in their current form only p-values are provided, without any descriptive statistics or confidence intervals to offer a better overall understanding for the readers.
  4. Line 229: Please change p=1.00 to p>0.99
  5. It is strongly recommended that Table 5 is modified in order to include descriptive statistics (mean, sd) of the variables and effect sizes (F/df of main effects and interactions).

Discussion

  1. In lines 347-348, the authors state that a serious limitation of their study is that they did not analyse the influence of the studied SNPs on suicidal behaviors. In spite of the limitation, it would be useful if the authors could provide descriptive data (frequency and relative frequency) of MDD patients in the sample that had demonstrated suicidal behavior.